# A Novel MDM2-Binding Chalcone Induces Apoptosis of Oral Squamous Cell Carcinoma

**DOI:** 10.3390/biomedicines11061711

**Published:** 2023-06-14

**Authors:** Guilherme Freimann Wermelinger, Lucas Rubini, Anna Carolina Carvalho da Fonseca, Gabriel Ouverney, Rafael P. R. F. de Oliveira, Acácio S. de Souza, Luana S. M. Forezi, Gabriel Limaverde-Sousa, Sergio Pinheiro, Bruno Kaufmann Robbs

**Affiliations:** 1Basic Science Department, Health Institute of Nova Friburgo, Fluminense Federal University, Nova Friburgo 28625-650, RJ, Brazil; guilhermefwermelinger@gmail.com (G.F.W.); lrubini@id.uff.br (L.R.); 2Postgraduate Program in Dentistry, Health Institute of Nova Friburgo, Fluminense Federal University, Nova Friburgo 28625-650, RJ, Brazil; fonseca.anna@gmail.com; 3Postgraduate Program in Applied Science for Health Products, Faculty of Pharmacy, Fluminense Federal University, Niteroi 24020-141, RJ, Brazil; ouverneygabriel@id.uff.br; 4Department of Organic Chemistry, Chemistry Institute, Fluminense Federal University, Niteroi 24020-141, RJ, Brazil; faelcarriello@gmail.com (R.P.R.F.d.O.); acacio.farma@gmail.com (A.S.d.S.); luanaforezi@id.uff.br (L.S.M.F.); 5Oswaldo Cruz Institute, Oswaldo Cruz Foundation, FIOCRUZ, Rio de Janeiro 21040-900, RJ, Brazil; gabriel.sousa@ioc.fiocruz.br

**Keywords:** chalcone, 1,2,3-triazole, MDM2, p53, antitumor, oral squamous cell carcinoma, apoptosis, cell cycle arrest, treatment

## Abstract

Oral squamous cell carcinoma (OSCC) represents ~90% of all oral cancers, being the eighth most common cancer in men. The overall 5-year survival rate is only 39% for metastatic cancers, and currently used chemotherapeutics can cause important side effects. Thus, there is an urgency in developing new and effective anti-cancer agents. As both chalcones and 1,2,3-triazoles are valuable pharmacophores/privileged structures in the search for anticancer compounds, in this work, new 1,2,3-triazole-chalcone hybrids were synthesized and evaluated against oral squamous cell carcinoma. By using different in silico, in vitro, and in vivo approaches, we demonstrated that compound **1f** has great cytotoxicity and selectivity against OSCC (higher than carboplatin and doxorubicin) and other cancer cells in addition to showing minimal toxicity in mice. Furthermore, we demonstrate that induced cell death occurs by apoptosis and cell cycle arrest at the G2/M phase. Moreover, we found that 1f has a potential affinity for MDM2 protein, similar to the known ligand nutlin-3, and presents a better selectivity, pharmacological profile, and potential to be orally absorbed and is not a substrate of Pg-P when compared to nutlin-3. Therefore, we conclude that 1f is a good lead for a new chemotherapeutic drug against OSCC and possibly other types of cancers.

## 1. Introduction

There are several types of cancers that affect the oral cavity; however, it is estimated that oral squamous cell carcinoma (OSCC) represents approximately 90% of all oral cancers. In addition, it is the eighth most common cancer in men [1]. Therefore, OSCC presents itself as a global public health problem due to its high incidence, with approximately 377,000 new cases and 177,000 deaths worldwide in 2020 [2,3].

The approaches used in the clinic for OSCC treatment take into account the stage of the disease, the site of onset, and the condition of the patient [4,5]. The most used chemotherapeutics include carboplatin, 5-fluorouracil, and cisplatin (Figure 1), alone or in combination. However, acute and chronic side effects can occur, including an increased probability of infections, bruising or bleeding, fatigue, and nerve and kidney damage [6]. Furthermore, the overall 5-year survival rate is 65% for OSCC, ranging from 84% for localized tumors to only 39% for metastatic cancers [1], mainly because diagnoses still occur in advanced stages of the disease [7]. Thus, it is necessary to develop new and effective anti-cancer agents that can improve the outcome of the disease and overcome toxicity.

A significant portion of the compounds used in cancer chemotherapy is derived from natural products [8,9,10]. Chalcones are biogenetic precursors of flavonoids and isoflavonoids and are the open-chain intermediates in the synthesis of aurones from flavones. Naturally occurring and synthetic chalcones have been shown to display a broad spectrum of biological activities, especially as anticancer agents [11,12,13]. Indeed, the naturally occurring chalcones licochalcones and flavokawain B (Figure 1) exhibited activity and induced apoptosis in human OSCC [14,15,16,17].

Modulation of the basic structure of chalcones by altering the aromatic residues leads to hybrid molecules generally with superior cytotoxic properties. In this context, the hybridization of chalcones with azoles is an important way to the development of novel anticancer agents [18,19,20]. Among them, some 1,2,3-triazole-chalcone hybrids exhibit remarkable toxicity against cancer cell lines acting through different mechanisms of action [18,21]. In some cases, the 1,2,3-triazole ring was proven to be a pharmacophore [22,23]. In recent years, we have reported natural flavonoids, terpenes, lignans [24,25,26,27], and also some synthetic naphthoquinones, as **Ia–c** (Figure 1), with significant activity against human OSCC [28,29,30,31].

Considering the concept of fragment-based drug design, we synthesized novel 1H-1,2,3-triazole-chalcone hybrids (**1**, Figure 1) by the combination of the natural chalcone flavokawain B and the naphthoquinones tethered to 1,2,3-triazoles **Ia–c** in an attempt to generate a synergistic cytotoxic effect, reduce side effects, and overcome drug resistance. The option for introducing the amino group is due to the strong cytotoxic effects on different cancer cell lines presented by 2′-aminochalcones [20,32,33].

This paper reports the first examples of 1H-1,2,3-triazole/chalcone hybrids as a novel class of potent cytotoxic compounds against OSCC. The possible mechanism of action and pharmacokinetic and toxicity parameters of the most promising derivative were investigated by employing in silico, in vitro, and in vivo approaches.

## 2. Materials and Methods

### 2.1. Chemistry 

#### 2.1.1. General Remarks

Reagents and solvents were used as purchased from commercial sources without further purification. Melting points were obtained on a Fisatom 430 digital melting-point apparatus (Fisatom, São Paulo, Brazil) and were uncorrected. The FT-IR spectra were obtained on a Perkin Elmer FT-IR Spectrometer Spectrum Two spectrometer (Perkin Elmer, Waltham, Massachusetts, EUA). 1H NMR spectra were recorded either on a Varian VNMRS at 500 MHz spectrometer or a Varian Unity Plus spectrometer at 300 MHz. 13C NMR-APT spectra were recorded on a Varian VNMRS at 125 MHz spectrometer or a Varian Unity Plus spectrometer at 75 MHz. Chemical shifts are reported in ppm and J values are given in Hertz. Signals are abbreviated as singlet, s; doublet, d; triplet, t; double-doublet, dd; double-double-doublet, ddd; quartet, q; multiplet, m.

#### 2.1.2. General Procedure for Production of 1,2,3-Triazole Alcohols

In a round-bottom flask equipped with a magnetic stirring bar, substituted aniline (10 mmol) was dissolved with 6N HCl (10 mL) in ice bath. A solution of NaNO^2^ (15 mmol) in 25 mL water was added dropwise, and the reaction mixture was stirred for 30 min. Then, sodium azide (40 mmol) dissolved in 50 mL water was added dropwise. After addition, the mixture was stirred for 2 h at room temperature. Then, the mixture was extracted with ethyl acetate (3 × 30 mL) and the combined organic extracts were washed with brine (3 × 30 mL), dried over anhydrous Na_2_SO_4_, filtered, and concentrated in vacuo. The residual crude azides were used directly without purification. A mixture of the appropriate aromatic azide (1 mmol), propargyl alcohol (1 mmol), CuSO^4^ pentahydrate (0.05 mmol), and sodium ascorbate (0.1 mmol) tert-butanol (7 mL) and H2O (7 mL) were stirred for 48–72 h at room temperature and subsequently extracted with ethyl acetate (3 × 30 mL). The combined organic extracts were washed with brine (3 × 30 mL), dried over anhydrous Na^2^SO^4^, filtered, and concentrated in vacuo. The residual crude product was purified via silica gel column chromatography using a gradient mixture of hexane and ethyl acetate to obtain the pure1,2,3-triazole alcohols.

##### Ethyl 4-(4-(hydroxymethyl)-1H-1,2,3-triazol-1-yl)benzoate

Yield: 84%. Yellow solid, mp 118–120 °C. IR (KBr, cm^−1^): 3228, 3094, 2979, 2942, 1702, 1604, 1267, 1068, 859, 784.1H NMR (500 MHz, CDCl3) δ ppm: 8.21 (2H, d, J = 8.9 Hz), 8.05 (1H, s), 7.83 (2H, d, J = 8.8 Hz), 4.91 (2H, s), 4.42 (2H, q, J = 7.1 Hz), 1.42 (3H, t, J = 7.1 Hz). 

##### (1-(2,6-Dimethylphenyl)-1H-1,2,3-triazol-4-yl)methanol

Yield: 84%. Orange oil. IR (neat, cm^−1^): 3205, 3132, 2919, 1483, 1378, 1201, 1042, 1021, 845, 775. 1H NMR (500 MHz, acetone-d6) δ ppm: 7.98 (1H, s); 7.36 (1H, t, J = 7.6 Hz), 7.26 (2H, d, J = 7.4 Hz), 4.79 (2H, s), 1.97 (6H, s). 

#### 2.1.3. General Procedure for Production of Compounds **2e** and **2f**

In a round-bottom flask equipped with a magnetic stirrer, freshly prepared manganese dioxide (150 mmol) and 10 mmol of the appropriate 1,2,3-triazole alcohol prepared in Section 2.1.2 were added to ethyl acetate (30 mL). The mixture was heated under reflux until all the triazole alcohol was consumed by TLC. The reaction mixture was filtered, and the filtrate was concentrated in vacuo to give the corresponding aldehydes **2e** and **2f** in satisfactory degrees of purity.

##### Ethyl 4-(4-Formyl-1H-1,2,3-triazol-1-yl)benzoate (**2e**)

Yield: 52%. Yellow solid, mp 105–107 °C. IR (KBr, cm^−1^): 3103, 2983, 1719, 1689, 1608, 1530, 1443, 1382, 1300, 1277, 1261, 1106, 853, 781. 1H NMR (500 MHz, CDCl3) δ ppm: 10.24 (1H, s), 8.60 (1H, s), 8.26 (2H, d, J = 8.7 Hz), 7.88 (2H, d, J = 8.7 Hz), 4.44 (2H, q, J = 7.1 Hz), 1.43 (3H, t, J = 7.1 Hz). 

##### 1-(2,6-Dimethylphenyl)-1H-1,2,3-triazole-4-carbaldehyde (**2f**)

Yield: 76%. Yellow solid, mp 92–94 °C. IR (KBr, cm^−1^): 3134, 2925, 2864, 1698, 1527, 1487, 1185, 1038, 771. 1H NMR (500 MHz, acetone-d6) δ ppm: 10.20 (1H, s), 8.83 (1H, s), 7.43 (1H, t, J = 7.6 Hz), 7.32 (2H, d, J = 7.6 Hz), 2.00 (6H, s). 

#### 2.1.4. General Procedure for Production of Compounds **1a–f**

2′-Aminoacetophenone 3 (1.35 g, 10 mmol) was added to a solution of the appropriate aldehyde **2a–f**, in 10 mL of ethanol with 200 mg of NaOH. After stirring at 0–5 °C for 12 h, the resulting solid was vacuum filtered and washed with ice water (3 × 30 mL). Solvent pair recrystallization was performed by solubilizing the crude 2′-aminochalcones in sufficient ethanol followed by filtration to remove solid impurities. To the solution was slowly added water until complete precipitation of 2′-aminochalcone. Filtration was followed by washing the solids with water (2 × 50 mL) and drying them in a desiccator.

##### (E)-1-(2-Aminophenyl)-3-(2-phenyl-2H-1,2,3-triazol-4-yl)prop-2-en-1-one (**1a**)

Yield: 84%. Yellow solid, mp 128–130 °C. IR (KBr, cm^−1^): 3447, 1654, 1615, 1577, 1500, 1243, 1150, 988, 759, 649. 1H NMR (500 MHz, DMSO-d6) δ ppm:8.63 (1H, s), 8.10–8.04 (3H, m), 7.99 (1H, d, J = 8.1 Hz), 7.65 (1H, d, J = 15.6 Hz), 7.58 (2H, t, J = 8.0 Hz; H), 7.44 (1H, t, J = 7.4 Hz), 7.37–7.25 (3H, m), 6.84 (1H, d, J = 8.4 Hz), 6.62 (1H, ddd, J = 8.1, 7.0 and 1.1 Hz). 13C NMR/APT (125 MHz, DMSO-d6) δ ppm: 189.66, 151.92, 145.88, 135.82, 134.24, 130.95, 130.80, 129.42, 128.88, 127.82, 127.19, 118.46, 117.09, 116.85, 114.38. 

##### (E)-1-(2-Aminophenyl)-3-(1-phenyl-1H-1,2,3-triazol-4-yl)prop-2-en-1-one (**1b**)

Yield: 55%. Yellow solid, mp 183–185 °C. IR (KBr, cm^−1^): 3457, 3334, 3134, 1651, 1614, 1595, 1499, 1465, 1445, 1282, 1258, 1155, 1045, 986, 768, 736. 1H NMR (300 MHz, DMSO-d6) δ ppm: 9.22 (1H, s), 7.99 (1H, d, J = 15.6 Hz); 7.95–7.86 (3H, m), 7.65 (1H, d, J = 15.5 Hz), 7.66–7.60 (2H, m), 7.55–7.49 (1H, m), 7.32–7.27 (3H, m), 6.84 (1H, dd, J = 8.4 and 1.1 Hz), 6.62 (1H, ddd, J = 8.1, 7.0 and 1.1 Hz). 13C NMR/APT (75 MHz, DMSO-d6) δ ppm: 189.9, 151.8, 144.4, 136.2, 134.1, 130.6, 129.8, 129.7, 128.7, 124.8, 122.9, 120.0, 117.3, 116.9, 114.4. 

##### (E)-1-(2-Aminophenyl)-3-(1-(4-methoxyphenyl)-1H-1,2,3-triazol-4-yl)prop-2-en-1-one (**1c**)

Yield: 94%. Yellow solid, mp 174–176 °C. IR (KBr, cm^−1^): 3408, 3310, 3121, 2837, 1652, 1615, 1581, 1513, 1481, 1449, 1257, 1181, 986, 845, 825, 767, 739. 1H NMR (300 MHz, DMSO-d6) δ ppm: 9.11 (1H, s), 7.97 (1H, d, J = 15.4 Hz), 7.92 (1H, dd, J = 4.4 and 1.4 Hz), 7.80 (2H, d, J = 9.0 Hz), 7.64 (1H, d, J = 15.6 Hz), 7.32–7.26 (3H, m), 7.16 (2H, d, J = 9.1 Hz), 6.83 (1H, dd, J = 8.4 and 1.0 Hz), 6.62 (1H, ddd, J = 8.2, 7.0 and 1.2 Hz); 3.85 (3H, s). 13C NMR/APT (75 MHz, DMSO-d6) δ ppm: 189.9, 159.4, 151.8, 144.2, 134.1, 130.6, 129.9, 129.6, 124.5, 122.8, 121.7, 117.3, 116.9, 114.8, 114.4, 55.4. 

##### (E)-1-(2-Aminophenyl)-3-(1-(4-nitrophenyl)-1H-1,2,3-triazol-4-yl)prop-2-en-1-one (**1d**)

Yield: 83%. Orange solid, mp 253–255 °C. IR (KBr, cm^−1^): 3129, 2922, 1682, 1596, 1537, 1507, 1369, 1343, 1260, 1211, 1010, 987, 851, 779, 749. 1H NMR (500 MHz, DMSO-d6) δ ppm: 9.41 (1H, s), 8.47 (2H, d, J = 9.1 Hz), 8.21 (2H, d, J = 9.1 Hz), 8.00 (1H, d, J = 15.6 Hz), 7.91 (1H, d, J = 8.1 Hz), 7.63 (1H, d, J = 15.6 Hz), 7.31–7.28 (3H, m), 6.83 (1H, d, J = 7.7 Hz), 6.62 (1H, ddd, J = 7.6, 6.8 and 0.9 Hz). 13C NMR/APT (125 MHz, DMSO-d6) δ ppm: 189.67, 151.84, 146.81, 144.86, 140.37, 134.17, 130.62, 129.17, 125.35, 125.30, 123.18, 120.57, 117.12, 116.90, 114.90, 114.40. 

##### Ethyl(E)-4-(4-(3-(2-aminophenyl)-3-oxoprop-1-en-1-yl)-1H-1,2,3-triazol-1-yl)benzoate (**1e**)

Yield: 59%. Yellow solid, mp 220–222 °C. IR (KBr, cm^−1^):3418, 3306, 3123, 2980, 1695, 1654, 1607, 1575, 1548, 1515, 1442, 1408, 1284, 1248, 1215, 1166, 1108, 1052, 984, 963, 859, 766, 740. 1H NMR (500 MHz, DMSO-d6) δ ppm: 9.34 (1H, s), 8.19 (2H, d, J = 8.8 Hz), 8.07 (2H, d, J = 8.8 Hz), 8.00 (1H, J = 15.6 Hz), 7.92 (1H, d, J = 7.3 Hz), 7.64 (1H, d, J = 15.6 Hz), 7.32–7.25 (3H, m), 6.83 (1H, d, J = 8.4 Hz), 6.67–6.58 (1H, ddd, J = 7.6, 6.9 and 1.0 Hz), 4.38 (2H, q, J = 7.1 Hz), 1.37 (3H, q, J = 7.1 Hz). 13C NMR/APT (125 MHz, DMSO-d6) δ ppm: 189.74, 164.57, 151.83, 144.67, 139.27, 134.14, 130.72, 130.64, 129.89, 129.42, 125.11, 122.91, 119.80, 117.17, 116.89, 114.40, 60.84, 13.85. 

##### (E)-1-(2-Aminophenyl)-3-(1-(2,6-dimethylphenyl)-1H-1,2,3-triazol-4-yl)prop-2-en-1-one (**1f**)

Yield: 55%. Orange solid, mp 175–177 °C. IR (KBr, cm^−1^): 3452, 3331, 3141, 2922, 2852, 1657, 1611, 1578, 1546, 1481, 1328, 1238, 1187, 1160, 1051, 1002, 973, 841, 786, 762. 1H NMR (500 MHz, DMSO-d6) δ ppm:8.78 (1H, s), 7.98 (1H, d, J = 15.6 Hz), 7.92 (1H, d, J = 8.2 Hz), 7.67 (1H, d, J = 15.6 Hz), 7.42 (1H, t, J = 7.6 Hz), 7.31 (2H, d, J = 7.6 Hz), 7.30–7.24 (3H, m), 6.83 (1H, dd, J = 8.4 and 0.9 Hz), 6.61 (1H, ddd, J = 8.1, 7.0 and 1.1 Hz), 1.98 (6H, s). 13C NMR/APT (125 MHz, DMSO-d6) δ ppm: 190.03, 151.79, 143.56, 135.30, 134.57, 134.11, 130.74, 129.97, 129.91, 126.82, 126.26, 124.49, 117.28, 116.89, 114.47, 16.64. 

### 2.2. Biological Assays

#### 2.2.1. Cells and Reagents

Human SCC-4, SCC-9, and SCC-25 cells derived from a human oral tongue SCC (squamous cell carcinoma) were obtained from the ATCC (CRL-1624, CRL-1629, and CRL-1628, respectively) and maintained in 1:1 DMEM/F12 (Dulbecco’s modified Eagle medium and Ham’s F12 medium; Gibco (Thermo Fisher, Waltham, MA, USA)) supplemented with 10% (*v*/*v*) FBS (fetal bovine serum; Invitrogen, Thermo Fisher, Waltham, MA, USA) and 400 ng/mL hydrocortisone (Sigma-Aldrich Co., St. Louis, MO, USA). Primary normal human gingival fibroblasts were obtained from the ATCC (PCS-201-018; HGF) and maintained in DMEM supplemented with 10% (*v*/*v*) FBS and were used in a maximum of six passages. The cells were grown in a humidified environment containing 5% CO_2_ at 37 °C. For all biological experiments, the compounds and control nutlin 3a were solubilized in 100% DMSO (all Sigma-Aldrich) to a final concentration of 10 mM. Carboplatin stock was prepared in water (Fauldcarbo^®^; LibbsFarmacêutica, São Paulo, SP, Brazil) and was used as a standard anticancer compound. 

#### 2.2.2. Cell Viability Assay (Cytotoxicity)

The viability of SCC cell lines, HT-29, HCT-116, HEP2G and primary human fibroblast cells was evaluated using the MTT assay as in [26]. Briefly, the cells were grown in duplicates in 96-well plates (5 × 10^3^ cells/well) until confluence. Then, the medium was removed, fresh medium was added, and the cells were returned to the incubator in the presence of different compounds. DMSO at the same concentrations was used as a 100% cell viability control. After 48 h, the cells were incubated with 5 mg/mL MTT reagent (3-(4,5-dimethyl-2-thiazolyl)-2,5-diphenyl-2-H-tetrazolium bromide) (Sigma-Aldrich Co., St. Louis, MO, USA) for 3.5 h. After that, formazan crystals were dissolved in MTT solvent solution (DMSO/methanol 1:1 *v*/*v*), and the absorbance at 560 nm was evaluated using an EPOCH microplate spectrophotometer (BioTek Instruments, Winooski, VT, USA) with the background absorbance at 670 nm subtracted. Each of the six compounds was tested at six or seven different concentrations, ranging from 0.3 µM to 200 µM in cancer cell lines (SCC-4, SCC-9 and SCC-25) and 0.3 µM to 200 µM in primary normal human gingival fibroblasts. Controls (carboplatin, nutlin 3a) were tested at six or seven different concentrations ranging from 0.05 µM to 1000 µM in cancer cells and normal cells, depending on the compound. 

#### 2.2.3. Hemolysis Assay

To determine the surfactant power of substances in biological membranes, a hemolysis assay was performed using human blood approved by the Research Ethics Committee of Universidade Federal Fluminense (CAAE: 43134721.4.0000.5626). Erythrocytes were collected by centrifugation at 1500 rpm for 15 min, washed with PBS (phosphate-buffered saline) supplemented with 10 mM glucose, and counted in an automatic cell counter (Thermo Fisher, Waltham, MA, USA). Erythrocytes were then plated in 96-well plates at a concentration of 4 × 10^8^ cells/well in triplicates, and 10 µL of compounds was added at a final concentration of 300 µM in PBS with glucose (final volume 100 µL). In total, 10 µL of PBS was used as a negative control and 10 µL of PBS with 0.1% Triton X-100 as a positive control. Data reading was performed with EPOCH (BioTek Instruments, Winooski, VT, USA) at an absorbance of 540 nm, and the statistical data were generated using the GRAPHPAD Prism 5.0 program (Intuitive Software for Science, San Diego, CA, USA).

#### 2.2.4. Cell Cycle and SubG1 Analysis

To evaluate the action of compound **1f** on the cell cycle and DNA fragmentation, SCC9 cell line cells were plated in a 6-well plate (5 × 10^5^ cells/well). After 48 h of treatment, the cells were trypsinized and stained with propidium iodide (75 µM) in the presence of NP-40. The DNA content was analyzed by collecting 10,000 events using a FACScalibur flow cytometer. The data were analyzed using CellQuest (BD Biosciences, Franklin Lakes, NJ, USA) and FlowJo (Tree Star Inc., Ashland, OR, USA) software as in [34].

#### 2.2.5. Apoptosis Analysis

Cells of the SCC9 cell line were plated in 6-well plates (5 × 10^5^ cells/well), trypsinized 48 h after treatment, labeled using the Annexin V-FITC Apoptosis Detection Kit according to the manufacturer’s protocol (#BMS500FI/300, Invitrogen), and analyzed by FACScalibur flow cytometry as in [35]. Furthermore, 5 × 10^4^ SCC-9 cells were plated in a 24-well plate containing 1 mL of DMEM/F12 with 10% FBS per well. CellEvent™ Caspase-3/7 Reagent (#R37111, Invitrogen) was diluted in a culture medium according to the manufacturer’s instructions. Twenty-four hours after plating, the cells were treated with Caspase-3/7 Reagent and 2 × IC_50_ of compound **1f** or DMSO as a control. The cells were analyzed by flow cytometry after 48 h of treatment.

#### 2.2.6. Statistical Analysis, IC_50_ Calculation

The data are presented as means ± SD. IC values for the MTT assays were obtained by nonlinear regression using the GRAPHPAD 5.0 program (Intuitive Software for Science, San Diego, CA, USA) from at least three independent experiments. A dose–response (inhibitor) vs. response curve using the least squares method was used to determine the IC_50_, SD, and R^2^ of the data. The selectivity index was calculated as SI = IC_50_ of the compound in normal oral fibroblast cells/IC_50_ of the same compound for each cancer cell line (SCC-4, SCC-9, SCC-25, HCT-116, HT29 and HEP2G), and the mean was calculated when indicated.

#### 2.2.7. In Vivo Acute Toxicity Study

The acute toxicity study for compound **1f** was performed as in the work of Macedo et al., 2019 [24]. The assay was approved by the University Animal Ethics Board under registration number 2699110419, in accordance with Brazilian guidelines and regulations. Dosing and analysis were performed according to OECD guidelines 423 and revised by Parasuraman [36]. The test was performed in 12-week-old C57BL/6 female mice via intraperitoneal injection. Each animal group had n = 3 and received only one intraperitoneal injection (Day 0) of compound **1f** dissolved in 3 mL PBS and 3% DMSO. The control group animals received only 3% DMSO in PBS. The first dose of the compound was 25 mg/kg; subsequent dose levels (50 mg/kg and 100 mg/kg) were determined based on the result obtained from the previous dosing. The animals were examined every day, twice a day, for mortality and morbidity for 14 days. At day 14, all animals were anesthetized (ketamine 100 mg/kg and xylazine 10 mg/kg) followed by cervical dislocation. The gross necropsy and histology of the main organs were performed. Body weight and average food consumption were measured every 7 days as an indication of morbidity, and the following signs were assessed: tremors; convulsion; salivation; diarrhea; lethargy; pain signs; increased rear arching; and defect in mobility. The necropsy included an examination of the external characteristics of the carcass; external body orifices; the abdominal, thoracic, and cranial cavities; and organs/tissues—liver, thymus, right kidney, right testicle, heart, and lung.

### 2.3. In Silico Studies

#### 2.3.1. Prediction of Toxicity and Pharmacokinetic Properties

The webserver SwissADME [37] was used to predict druglikeness and pharmacokinetic behavior based on molecular structure. Briefly, the structure in SMILES format was used as input, and the output is a series of parameters including Lipinski rule of five [38], providing an insight into pharmacokinetics. The values of the calculated octanol–water partition coefficient (cLogP), molecular weight (MW), number of hydrogen bond acceptors (nON), number of hydrogen bond donors (nOH/NH), and topological polar surface area (TPSA) were calculated. The novel chalcones were summited for characterization, and nutlin-3a, the well characterized MDM2 inhibitor, and doxorubicin, a well-established drug used in treatment of an array of different cancers, including OSCC, were also submitted as reference. 

#### 2.3.2. In Silico Docking Studies

The crystallographic structure of MDM2 associated with nutlin-3a (PDB entry 4HG7) was chosen for molecular docking studies [39]. The novel chalcones were built in silico using Avogadro v1.2 [40] and then inserted as input into the PRODRG [41] server for generation of molecular topology and coordinates. Docking studies were carried out using AutoDock 4 with AutoDockTools [42] and 500 runs of Lamarckian genetic algorithm. Other parameters were kept default. The grid box was built based on the docking site of nutlin-3a in Mdm2 and its respective pocket, resulting in a box with dimensions 43 × 64 × 46 Å (X,Y,Z), centered on the coordinates −24.417, 9.389, −10.167 (X,Y,Z). A redock test was performed using Mdm2, and nutlin-3a as a ligand. Results presented a RMSD of 0.7 Å, therefore validating the chosen method on predicting dock in this specific system. Chalcones were tested as ligands using the same grid box defined on the redocking process, following the same parameters and protocol.

#### 2.3.3. Molecular Dynamics Calculation

To verify complex stability, the best docking generated complex and the reference system MDM-2/nutlin-3a were submitted to 100 ns molecular dynamics calculations using GROMACS [43]. Before the simulations, the structure of MDM2 (PDB entry 4HG7) was mutated back to its wild-type sequence to avoid any simulation artifacts caused by the mutations introduced by the authors to facilitate protein crystallization [39]. Ligand topologies were generated via the webserver PolyParGen [44] for AMBER99SB force field [45]. The system was solvated in TIP3P water, adding 4 Cl-ions as necessary for neutralization of the system net charge. Energy minimization and equilibration phases were performed, preparing the system for the simulation. Analyses were performed using the GROMACS package tools and our “in house” software SurfInMD based on the Connolly surface [46] [Limaverde-Sousa et al., manuscript in preparation]. Cluster analysis was performed using GROMOS method with 2 Å cutoff with at least 100 frames. Images were generated using ChimeraX and PyMOL [47,48].

## 3. Results and Discussion

### 3.1. Chemistry

While the 1,2,3-triazole aldehydes **2a–d** have been previously described in the literature (Figure 2) [21,49] compounds **2e** and **2f** were prepared by oxidation of the corresponding alcohols according to classic synthetic route Methods [50].

The Claisen–Schmidt condensation of the commercially available 2-aminoacetophenone **3** with the 1,2,3-triazole aldehydes **2a–f** produced the respective novel (*E*)-1,2,3-triazole chalcones **1a–f** in moderate to good yields after recrystallization from EtOH/H_2_O (Figure 2) [21]. All chemical compounds spectra are in the Appendix A.

### 3.2. Biological Assays

#### 3.2.1. Cytotoxicity, Selectivity, and Hemolytic and Toxic Potential of New Chalcones

Initially, the six chalcones (**1a–f**) were submitted to the MTT assay to evaluate their cytotoxicity. The assay was performed using the oral cancer cell line SCC9, and the results were analyzed by a non-linear regression curve to determine the value of the half maximal inhibitory concentration (IC_50_). As controls, known chemotherapeutic agents were used, namely carboplatin, routinely used in the treatment of oral cancer, and doxorubicin, widely used for other types of cancers.

Of the six chalcones tested, all presented dose-dependent cytotoxicity, except **1d** and **1e** (Table 1). Although chalcones **1a** and **1c** displayed high cytotoxicity against SCC9 (IC_50_ of 9.95 and 9.32 µM), both formed highly insoluble well-structured crystals at low concentrations, making it impossible to accurately test their cytotoxicity in vitro and indicating them as poor drug candidates. These compounds were excluded from further biological analyses. On the other hand, compounds **1b** and **1f**, reported here for the first time, were highly soluble and showed noteworthy anticancer activities with IC_50_ of 12.72 and 3.87 µM, respectively, significantly lower than carboplatin (IC_50_ = 155.67 µM) and similar to doxorubicin (IC_50_ = 2.99 µM; Table 1). 

The initial screening was performed in SCC9 cells because it is usually more sensitive to anticancer drugs [31]. However, to restrict possible abnormalities in the behavior of a single cell line when compared with cancer cells in patients it is important to consider other oral cancer cell lines, as well as determining whether the effect is general or cancer specific. Therefore, the cytotoxicity of selected substances was tested in two additional SCC tumor cell lines (SCC4 and SCC25; Table 2) and also on cancer cell lines from different origins (Table 3). The IC_50_ of **1b** and **1f** for SCC4 and SCC25 were very low and similar to what was found for SCC9, proving their robust cytotoxic effect of these compounds on OSCC cells. 

The degree of selectivity of a molecule is expressed by its selectivity index (SI). When a substance presents SI ≥ 2, it is selectively more toxic for cancer cells; a value of SI < 2 indicates cytotoxicity for normal cells [28,29,51]. For the SI determination, primary human gingival fibroblasts (HGF) were used. For each compound, the SI value was calculated using the given formula: SI = IC_50_ normal cell/IC_50_ cancer cells (Table 2 and Table 3). Among the evaluated compounds, chalcone 1f was highly selective against the OSCC cell lines tested, with an average SI value of 6.51 (Table 2). Furthermore, compound **1f** was highly selective among other cancer cell types with an SI value of 7.55 against colorectal adenocarcinoma (HT29; Table 3) although slightly below the selectivity for SCC9 cell line (SI = 7.63; Table 2). 

Altogether, compound **1f** stands out as a potent cytotoxic and selective against three different OSCC and other cancer cells, being even more efficient than the controls used in clinic, carboplatin and doxorubicin. In the future it will be interesting to validate these results using in vivo tumor models as xenograf of OSCC cells in immunodeficient mice [8] or 3D culture models [52,53].

Since **1f** was the most cytotoxic and selective compound tested, we proceeded to verify its potential for clinical application. To verify the surfactant activity of the compound on cell membranes, a hemolysis test was performed. Figure 2 shows that compound **1f** lack hemolytic potential, with less than 2% hemolysis compared to the positive control, Triton X-100, which represents 100% of lysis in red blood cells. This result rule out nonspecific cytotoxicity through cell membrane damage and enable following in vivo assays.

Pre-clinical tests in animals are a very important step for drug development and for understanding the therapeutic potential of new molecules [38]. With the absence of hemolytic activity, we started the acute toxicity tests. Assessing acute toxicity involves administering a single dose of a substance or extract to a particular species, which enables identification of the toxic impact on specific organ, dose, and species under scrutiny [38]. The evaluation of toxicity is an essential element in the process of developing new pharmaceuticals. The acute toxicity assay involved the intraperitoneal administration of **1f** to C56BL/6 mice, and the animals were monitored for a period of 14 days. Throughout the experiment, food consumption, animal weight, and any observed morbidities were documented, and morphological and pathological changes were studied. The initial concentration of **1f** used was 25 mg/kg, but due to the absence of morbidity and mortality, subsequent groups were given higher doses of 50 mg/kg and 100 mg/kg. During the first week of the experiment, all treated groups showed a reduction in food intake, but it was more significant in the group treated with 100 mg/kg (as indicated in Figure 3B). In the second week, there was no noticeable difference in food consumption between all groups. Despite this reduction in food intake, there was no significant change in animal weight when compared to the control group (as shown in Figure 3C). Furthermore, no morbidities, mortalities, or macroscopic changes were observed during analysis and necropsy as depicted in Appendix A. The findings indicate that the concentrations of **1f** tested had minimal toxicity in C56BL/6 mice, thereby making it a suitable candidate for in vivo anticancer trials.

#### 3.2.2. Prediction of Anticancer Target of **1f** by Molecular Docking and Modeling

The literature was reviewed to find potential targets in which the novel chalcones could fit as ligands. Several different natural or semi-synthetic chalcones have shown antitumor activity due to the inhibition of some molecular targets such as mTOR, B-Raf, NF-κB, topoisomerase-II, the JAK/STAT signaling pathways and others [54], and more recently at the MDM2/p53 pathway [55]. 

MDM2 is an oncogenic protein related to the physiological regulation of p53 [56]. The overexpression of murine double minute 2 (MDM2) is a recurring alteration that contributes to the survival of tumors that express p53 wild-type proteins, protecting transformed cells against p53 pathway mediated apoptosis, representing an interesting target for drug development. In tumorigenesis, MDM2 promotes degradation of p53, preventing it from exercising its antiproliferative functions [57]. Amplification of the *MDM2* gene is observed in several tumors, including OSCC [58]. The nutlins were one of the first compounds identified to act on the MDM2/p53 interaction; it was observed that they are able to displace p53 from MDM2 in vitro [59].

Since MDM2 has been previously reported as a possible target for chalcones [54,60], and considering its importance in cancer development and progression, we proceeded to a detailed analysis by reverse docking for all cytotoxic chalcones (**1a**, **1b**, **1c** and **1f**) using nutlin-3a as control.

The redock of nutlin-3a to MDM2 was successful, with minimal conformational changes (RMSD of 0.7 Å) presenting a binding energy of −9.52 kcal/mol. Compound **1f** presented a binding energy of −9.18 kcal/mol, the lowest among the chalcones in this study (Appendix A), which correlates to the lowest IC_50_ obtained on in vitro studies (Table 1). Interactions for all four cytotoxic chalcones were mapped using Discovery Studio (Appendix A), and those specifically for **1f** are represented at Figure 3A. The most prevalent docking cluster of compound **1f** with MDM2 shows the ligand with a bent conformation, making an intramolecular CH–π interaction between one of the two methyl groups of the dimethylphenyl ring and the ring of the chalcone group and binding to the hydrophobic pocket of MDM2, contacting residues L54, V93, F91, I99. The chalcone amino group of **1f** also makes hydrogen bonds with the mainchain of Q24 and sidechain of Y100, making it a possible good ligand for MDM2 protein.

In recent years, several nutlin derivatives have entered clinical trials for the treatment of various cancers, including leukemia, lymphoma, and solid tumors [61]. Another nutlin derivative, AMG-232, has shown activity in preclinical studies and is currently being evaluated in clinical trials for multiple solid tumors [61]. Although the promising results for nutlin and possibly other MDM2 inhibitors, studies have reported the development of nutlin resistance in some cancer cells, which may also limit their long-term clinical use [62]. Additionally, nutlins may cause adverse effects, such as gastrointestinal symptoms, hematological abnormalities, and hepatic impairment [63]. Thus, the development of new, less toxic MDM2 inhibitors is a subject of paramount importance.

Based on the docking results with compound **1f**, which showed similar binding energy to the control (the well-known MDM2 inhibitor nutlin-3a), we further analyzed the stability of the complex MDM2-**1f** by molecular dynamics calculations in comparison to the control complex (MDM2-nutlin-3a). In contrast to the stable interaction of nutlin-3a with MDM2, which maintains the contacts observed in the crystallographic structure during molecular dynamics, a significant rearrangement was observed for the complex MDM2-**1f**, with the formation of a set of clusters as shown by Figure 3B. The starting conformation of the residues that form the cavity of MDM2 (originally from the crystallographic structure with nutlin-3a) is disturbed by the absence of the nutlin-3a. A fast induced fit of the pocket is observed on the first 5 ns to adapt the cavity conformation to interact with compound **1f**. After 5 ns, the predominant cluster takes place, assuming the highest average interface area and lowest interaction energy during the simulation Figure 3C,D. The compound **1f** assumes an extended configuration, mainly interacting with hydrophobic residues (Figure 3E,F). The dimethylphenyl group makes contacts with residues L54, F86, F91, H96 and I99, the triazole ring with residues L57, G58, V73, V75, V93, and H96 and the chalcone group with residues I61, M62, and Y67. Within the last 10 ns of the molecular dynamics trajectory, the amine group of the chalcone establishes a hydrogen bond with the main chain with Q72, as observed by the stabilization of the coulombic component of the interaction energy around −30 kcal/mol (Figure 3D,G).

The repeated formation of the same preferential cluster during molecular dynamics with no ligand dissociation indicates that the interaction between compound **1f** and MDM2 is feasible, although a larger sampling may be needed for the stabilization of the complex. This result may also be indicative that the dynamic nature of the binding implies low residency time of this chalcone as an Mdm2 inhibitor, although potentially sufficient to disrupt the binding of this protein with other partners.

#### 3.2.3. Prediction of Toxicity and Pharmacokinetic Properties of Compound **1f**

In the past decade, approximately 50% of drug candidates failed due to absorption, distribution, metabolism, excretion, and toxicity, collectively known as ADMET parameters [64]. Computational pharmacology employs in silico assays that can predict and infer how drugs impact biological systems, ultimately improving drug development and preventing unwanted side effects [65]. To assess the oral bioavailability of the new chalcones compounds, relevant parameters were computed and compared to clinical approved compounds (carboplatin and doxorubicin) and with nutlin-3a using the SwissADME servers. Lipinski’s “rule of 5” was employed to evaluate oral bioavailability based on four criteria: (1) the logarithm of the octanol/water partition coefficient (cLogP) ≤ 5; (2) the number of hydrogen bond acceptors (nON) ≤ 10; (3) the number of hydrogen bond donors (nOH/NH) ≤ 5; and (4) molecular weight (MW) ≤ 500 Da [40]. Compounds that violate two or more of these criteria likely exhibit inadequate permeation and absorption. Chalcones **1f** adhered to all Lipinski’s “rule of 5,” while nutlin-3a, doxorubicin, and carboplatin had 1,3, and zero violations, respectively (Table 4 and Appendix A). Furthermore, the topological polar surface area (TPSA) is a parameter used in predicting drug cell permeability, oral bioavailability, and intestinal absorption. A TPSA above 140 Å^2^ indicates low membrane permeability, whereas a TPSA below 60 Å^2^ suggests high permeability and human intestinal absorption [66]. Based on the values presented in Table 4, compound **1f** has an intermediate TPSA value (114.26 Å^2^) higher then nutlin-3a but lower than carboplatin (126.6 Å^2^) and doxorubicin (206.1 Å^2^), indicating favorable cell permeability.

To strengthen the rule-based prediction of absorption and permeability, a QSAR-based method available within the admetSAR 2.0 server was used to predict the bioavailability of compound (Table 4, Appendix A). The results showed that compound **1f** is predicted to have good oral bioavailability, comparable to nutlin-3a, while the drugs doxorubicin and carboplatin were predicted to have poor oral bioavailability. This is in line with experimental studies that have demonstrated the low oral bioavailability of these drugs and the need for intravenous administration [67,68], validating the predictions made in this study. This supports the suitability of compound **1f** for oral delivery, unlike the evaluated anticancer drugs. Given that phosphoglycoprotein-P (Pg-P) is associated with drug resistance, the server evaluated whether compound **1f** could act as a substrate or inhibitor of this protein. The results showed that compound **1f** is not predicted to be a substrate or inhibitor of Pg-P (Table 4).

Interesting, nutlin-3a seems to be both substrate and a possible inhibitor of Pg-p protein, possibly accounting for the drug resistance phenotype associated with its clinical use [62]. Similarly, carboplatin was not predicted to act as a substrate or inhibitor of Pg-P, while doxorubicin was predicted to be a substrate but not an inhibitor of this protein, implying its transportation and expulsion through efflux pumps. These predictions are consistent with available experimental data for both drugs [69]. Therefore, the in silico analyses suggest that compound **1f**, in addition to having a good pharmacological profile, has the potential to be orally absorbed and is not a substrate of Pg-P, increasing its likelihood as a promising drug candidate.

Corroborating the good pharmacological properties, molecular docking and modeling, compound **1f** displayed higher cytotoxicity and selectivity against SCC9 and HT29 cells when compared to nutlin-3a (Table 5). Therefore, we conclude that **1f** is a good lead for a new chemotherapeutic drug against OSCC and possible other types of cancers.

#### 3.2.4. Cell Death Investigation

Considering the findings that indicate the selectivity and tolerability of compound **1f** in mice, our attention shifted to identifying the potential mechanism and pathway involved in cell death. Various types of cell death can be triggered by chemotherapy, and identifying the precise pathway is crucial for the advancement of new anticancer drugs [70].

Since MDM2 inhibition led to p53 activation and cell death through apoptosis and cell cycle arrest, we investigated these events by flow cytometry analysis. The treatment of SCC9 cells with compound **1f** after 48 h showed a significant increase in phosphatidylserine exposure (Figure 4A, **1f**: 43.0% vs. DMSO: 4.4%), fragmentation of DNA (Figure 4B, 1f: 23.3% vs. DMSO: 2.2%), caspase 3/7 expression (Figure 4C, **1f**: 47.5% vs. DMSO: 14.9%) and an arrest in the G2/M phase of cell cycle (Figure 4D, **1f**: 44.0% vs. DMSO: 28.7%). All of these are hallmarks of apoptosis [71], indicating this is the type of cell death occurring.

Corroborating our data, nutlin-3a and other MDM2 inhibitors can induce apoptosis and cell cycle arrest at G2/M [72,73,74] with promising applications to the clinic as cytotoxic chemotherapy [75]. Further, different chalcones can induce apoptosis by a caspase-dependent mechanism through the intrinsic pathway and arrest at G2/M phase of cell cycle [76,77]. Altogether, the results demonstrate that compound **1f** promotes oral cancer cell death by apoptosis and cell cycle arrest at G2/M.

## 4. Conclusions

A series of novel 1,2,3-triazole-chalcone hybrids was easily synthesized by the Claisen–Schmidt classic methodology and evaluated against oral squamous cell carcinoma. The novel 1H-1,2,3-triazole-chalcone hybrid **1f** demonstrated great cytotoxicity and selectivity against OSCC and other cancer cells in vitro, besides showing minimal toxicity in mice, making it a suitable candidate for in vivo anticancer trials. Furthermore, flow cytometry studies proved compound **1f** induces death by apoptosis and cell cycle arrest at G2/M phase. In fact, in silico investigations found that **1f** has a potentially affinity for MDM2, an oncogenic protein that regulates p53 activity, giving a possible mechanistic explanation for the cell death observed. Moreover, **1f** was shown to present a good pharmacological profile, the potential to be orally absorbed and is not a substrate of Pg-P, a protein associated with drug resistance. Therefore, we conclude that **1f** is a good lead for a new chemotherapeutic drug against OSCC and possibly other types of cancers. As strengths, we show an easy, cheap synthetic protocol to generate a novel chalcone that has strong cytotoxicity and selectivity in vitro against different types of cancers, has low toxicity in vivo, and has good pharmacokinetic properties with a possible binding capability to MDM2 protein. In the future, it will be interesting to address the antitumoral effect in in vivo or 3D culture tumor models and verify the direct biding of this chalcone and the disruption of the MDM2/p53 pathway.

## Data Availability

Data will be available upon request.

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
