# Peer review of "A Novel MDM2-Binding Chalcone Induces Apoptosis of Oral Squamous Cell Carcinoma"

_biomedicines, 2023, doi:10.3390/biomedicines11061711_

Round 1
Reviewer 1 Report
The authors developed several chemicals and evaluated their selectivity index to find the tumor-specific agent with low cytotoxicity to normal cells.
1. The main question addressed by the research is that these authors provide a well-designed and well-written work for providing new chemotherapeutic drug against OSCC and possible other types of cancers.
2. The topic is original and relevant in the suitable field for this special issue.
3. Compared with other published material, the authors performed the cell cycle and apoptosis assays to validate the G2/M arrest and apoptotic effects to oral cancer cells and others. They also conducted the molecular docking to suggest that MDM2 is a potential target for compound 1f.
4. The specific improvements are that they have synthesized several chemical compounds and assessed their cytotoxicity and apoptosis. They have used the Carboplatin and Doxorubicin as the IC50 control.
5. The conclusions consistent with the evidence and arguments presented
and do they address the main question posed.
6. The references appropriate.
7. No any additional comments on the tables and figures.
Minor comments:
1. Please add the name for primary human fibroblast cells, i.e. HGF.
Author Response
Response:
We thank the reviewer for the positive comments. We added the name of the ATCC Gengival Primary Human fibroblast (PCS-201-018; HGF) to the methods, the results and tables.
Reviewer 2 Report
The study highlights an important therapeutic aspect of treating oral squamous cell carcinoma (OSCC). Utili 1,2,3-triazole-chalcone hybrids against OSCC is a smart alternative approach. However, there are some issues required to be addressed before further consideration.
Please separate the results and discussion into two different segments.
Please add the strengths and limitations of the current study.
The study is based on different in silico, in vitro, and in vivo approaches. However, it could be further explored using tumor spheroids embedded in an extracellular matrix or tumor microenvironment matrices. In the discussion and future part, please add a section that these results should be further validated using 3D culture models where the cancer cells will form spheroids in such matrices which will further suit the human microenvironment-mimicking matrices. (https://doi.org/10.1016/j.yexcr.2018.06.037 ; https://doi.org/10.1186/s12885-015-1944-z ). I recommend using the references as the data was analyzed using animal and human-microenvironment-based 3D tissue models.
Author Response
Minor comments:
The study highlights an important therapeutic aspect of treating oral squamous cell carcinoma (OSCC). Utili 1,2,3-triazole-chalcone hybrids against OSCC is a smart alternative approach. However, there are some issues required to be addressed before further consideration.
1) Please separate the results and discussion into two different segments.
Response:
We thank the reviewer for the suggestion. Although is a good suggestion we think that for this kind of paper its more interesting to make a more concise and direct discussion together with the results (Results and Discussion). While some journals have a mandatory separation between this two section, Biomedicines (and MDPI journal overall) do not make this an obligation, as described at the Authors Instruction (https://www.mdpi.com/journal/biomedicines/instructions#preparation): “Discussion: Authors should discuss the results and how they can be interpreted in perspective of previous studies and of the working hypotheses. The findings and their implications should be discussed in the broadest context possible and limitations of the work highlighted. Future research directions may also be mentioned. This section may be combined with Results.”
2) Please add the strengths and limitations of the current study.
Response:
We added the following text to the conclusion: “As strengths we show and easy, cheap synthetic protocol to generate a novel chalcone that has a strong cytotoxicity and selectivity in vitro against different types of cancers, low toxicity in vivo, and good pharmacokinetic properties with a possible binding capability to MDM2 protein. In the future it will be interesting to address the antitumoral effect in a in vivo or in 3D culture tumor models and verify the direct biding of this chalcone and the disruption of the MDM2/p53 pathway.”
3) The study is based on different in silico, in vitro, and in vivo approaches. However, it could be further explored using tumor spheroids embedded in an extracellular matrix or tumor microenvironment matrices. In the discussion and future part, please add a section that these results should be further validated using 3D culture models where the cancer cells will form spheroids in such matrices which will further suit the human microenvironment-mimicking matrices. (https://doi.org/10.1016/j.yexcr.2018.06.037 ; https://doi.org/10.1186/s12885-015-1944-z ). I recommend using the references as the data was analyzed using animal and human-microenvironment-based 3D tissue models.
Response:
We thank the reviewer for the suggestion. We added the following text to the results and the abovementioned text in the conclusion: “In the future it will be interesting to validate these results using in vivo tumor models as xenograf of OSCC cells in immunodeficient mice (de Queiroz et al., 2023) or 3D culture models (Hoque Apu et al., 2018; Salo et al., 2015).”